# High-resolution comparative atomic structures of two Giardiavirus prototypes infecting *G. duodenalis* parasite

Han Wang[1], Gianluca Marucci[2], Anna Munke[1,3], Mohammad Maruf Hassan[1], Marco Lalle[2]*, Kenta Okamoto[1]*

1 The Laboratory of Molecular Biophysics, Department of Cell and Molecular Biology, Uppsala University, Uppsala, Sweden, 2 Unit of Foodborne and Neglected Parasitic Diseases, Department of Infectious Diseases, Istituto Superiore di Sanità (ISS), Rome, Italy, 3 Center for Free-Electron Laser Science CFEL, Deutsches Elektronen-Synchrotron DESY, Hamburg, Germany

* marco.lalle@iss.it (ML); kenta.okamoto@icm.uu.se (KO)

## Abstract

The Giardia lamblia virus (GLV) is a non-enveloped icosahedral dsRNA and endosymbiont virus that infects the zoonotic protozoan parasite *Giardia duodenalis* (syn. *G. lamblia*, *G. intestinalis*), which is a pathogen of mammals, including humans. Elucidating the transmission mechanism of GLV is crucial for gaining an in-depth understanding of the virulence of the virus in *G. duodenalis*. GLV belongs to the family *Totiviridae*, which infects yeast and protozoa intracellularly; however, it also transmits extracellularly, similar to the phylogenetically, distantly related toti-like viruses that infect multicellular hosts. The GLV capsid structure is extensively involved in the longstanding discussion concerning extracellular transmission in *Totiviridae* and toti-like viruses. Hence, this study constructed the first high-resolution comparative atomic models of two GLV strains, namely GLV-HP and GLV-CAT, which showed different intracellular localization and virulence phenotypes, using cryogenic electron microscopy single-particle analysis. The atomic models of the GLV capsids presented swapped C-terminal extensions, extra surface loops, and a lack of cap-snatching pockets, similar to those of toti-like viruses. However, their open pores and absence of the extra crown protein resemble those of other yeast and protozoan *Totiviridae* viruses, demonstrating the essential structures for extracellular cell-to-cell transmission. The structural comparison between GLV-HP and GLV-CAT indicates the first evidence of critical structural motifs for the transmission and virulence of GLV in *G. duodenalis*.

## Author summary

We determined the first atomic structure of a unique dsRNA virus known as the Giardia lamblia virus (GLV). The structure of GLV is important for two reasons. First, the GLV structure is the representative structure of a primitive group of viruses that infects unicellular parasites but has adapted an extracellular lifestyle. Hence, by comparing the structure of GLV with that of other dsRNA viruses, we could identify essential surface loop

**Data Availability Statement:** The cryo-EM maps of GLV-HP and GLV-CAT are available in the EMDB database, see entries EMD-18791 and EMD-18792, and the atomic models of the CPs are available in the PDB database, see entries 8R0F and 8R0G. All

other relevant data are within the manuscript and its Supporting information files.

**Funding:** This work was supported by the Swedish Research Council (Vetenskapsrådet, grant numbers 2018-03387 and 2023-01857 to KO and, grant number 2022-00236 to AM, https://www.vr.se/english.html); by the Swedish Foundation for International Cooperation in Research and Higher Education (STINT) (grant number JA2014-5721 to KO, https://www.stint.se); by FORMAS research grant from the Swedish Research Council, for Environment, Agricultural Sciences and Spatial Planning (Svenska Forskningsrådet Formas, grant number 2018-00421 to KO, https://formas.se/); by the Royal Swedish Academy of Sciences (grant number BS2018-0053 to KO, https://www.kva.se/en/); by The Research Council of Norway (Norges Forskningsråd, grant number 324266 to KO, https://www.forskningsradet.no/en/); by the Istituto Superiore di Sanità (grant number ISS20-4389733b36a1 to ML, https://www.iss.it/). The funders had no role in study design, data collection and analysis, decision to publish, or preparation of the manuscript.

**Competing interests:** The authors have declared that no competing interests exist.

structures that have evolved to enable dsRNA viruses to infect their hosts extracellularly. Second, the GLV structure empowers us to formulate a strategy for engineering GLV, with a focus on its future therapeutic applications. *G. duodenalis*, a common zoonotic parasite, can infect both humans and domestic animals, often leading to severe diarrhea. Conventional anti-parasitic drugs are employed to treat *G. duodenalis* symptomatic infections; however, cases of refractories to these drugs are increasingly being reported. A new alternative approach gaining interest is virotherapy, which harnesses viruses that infect parasites to combat infections. An in-depth structural comparison between two GLV prototypes revealed significant structural mechanisms that potentially enhance the virulence of GLV for efficiently clearing Giardia parasites from infected patients.

## Introduction

*Giardia duodenalis* (syn. *G. lamblia* and *G. intestinalis*) is a zoonotic intestinal protozoan parasite that infects the upper part of the small intestine of mammals, including humans. *G. duodenalis* causes giardiasis, a widespread diarrheal disease in humans [1]. The pathogen is classified into eight distinct genetic groups or Assemblages (A–H), with human infection almost exclusively associated with Assemblages A and B, whereas giardiasis in animals is due to host-specific Assemblages (C–H). Assemblages A and B also have zoonotic potential, having been isolated from both humans and animals infected with *G. duodenalis* [2]. Transmission of *G. duodenalis* occurs through the fecal-oral route by accidental ingestion of cysts, the infective environmental resistant stage of the parasite, by direct contact with stools of humans and animals infected with the parasite or by drinking water or eating food (e.g., fresh produce) contaminated with the parasite's cysts. Outbreaks of giardiasis are frequently reported globally [3,4] as well as recently in EU countries in 2018–2019 [5,6]. However, infection with *G. duodenalis* is still considered a neglected disease; thus, effective anti-parasitic drugs are limited [7], vaccines are not yet available [8,9], and treatment failure with nitroimidazoles, the first-line drug for giardiasis, is increasingly reported, with up to 45% of patients not responding to initial treatment [10].

For neglected human protozoan parasitic infections, virotherapy using parasite-specific, endosymbiont viruses has been proposed as an alternative/integrative approach to control them [11–13]. The double-stranded (ds)RNA virus Giardia lamblia virus (GLV or Giardiavirus) exclusively infects *G. duodenalis* [14,15]. GLV belongs to the genus *Giardiavirus* in the family *Totiviridae*. Other *Totiviridae* viruses include diverse dsRNA viruses that exclusively infect yeasts and fungi, such as the Saccharomyces cerevisiae virus L-A (ScV-L-A), the Saccharomyces cerevisiae virus L-BC (ScV-L-BC), and the Helminthosporium victoriae virus 190S (HvV190S), as well as those that infect protists, such as the Trichomonas vaginalis virus (TVV) (genus *Trichomonasvirus*) and Leishmania spp. RNA virus (LRV) (genus *Leishmanivirus*) [15–17]. The genetically distantly related toti-like viruses, for example, mosquito Omono River virus (OmRV), shrimp infectious myonecrosis virus (IMNV), and salmon piscine myocarditis virus (PMCV), infect a broad range of invertebrates and vertebrates [16,18,19].

GLV is phylogenetically close to toti-like viruses compared to other *Totiviridae* viruses, which hints at an understanding of the evolutionary relevance of *Totiviridae* viruses and toti-like viruses [16,20]. Almost all *Totiviridae* viruses infecting unicellular yeast and protists merely adopt intracellular transmission by frequent mating of cell division and cell fusion in yeasts and protists [17,21,22]. In contrast, the toti-like viruses infecting multicellular hosts have employed an extracellular phase and likely transmit adjacent cells using membrane

penetration and/or receptor-binding mechanisms [23–26]. Intriguingly, although GLV infects a unicellular protozoan, it can be efficiently transmitted extracellularly [15,27].

*Totiviridae* and toti-like viruses encode two open reading frames (ORFs), ORF1 and ORF2, in their genome [15,16]. ORF1 expresses a capsid protein (CP) that assembles an icosahedral capsid. Only for the toti-like viruses is the CP further post-translationally processed to form major CP (MCP) and crown protein (CrP) [16,24]. These viral CP (or MCP) and CrP are in charge of the infection of their hosts [24,28]; however, their role in infection is still poorly understood. The CP of *Totiviridae* viruses has another function: snatching the cap RNA structure of the host. The attachment of the 5'-cap structure (7-methylguanosine linked through a reverse 5'-5' triphosphate bridge; 5'-m$^7$GpppX) to mRNA is required for effective translation in eukaryotic cells. Many viruses express enzymes, such as RNA triphosphatase, guanylyltransferase, and methyltransferase, to process viral RNA via several catalytic pathways and create the 5'-cap structure; however, some viruses, especially negative sense single-stranded ((-)ss) RNA viruses, use a cap-snatching approach and utilize a short cleaved capped RNA fragment from the host mRNAs [29,30]. The invariant His residue of the CP in the yeast *Totiviridae* viruses binds covalently to the m$^7$GpppG cap structure of the host mRNA, thereby decapping the m$^7$Gp moiety, which speculates a unique cap-snatching approach of transferring the m$^7$Gp to the 5′ end of the synthesized viral positive sense (+)ssRNA transcripts [31,32]. However, for GLV, the transcripts lack the 5'-cap structure, and a cap-independent internal ribosomal entry site (IRES) could promote the translation of the structure [15,33].

ORF2 encodes RNA-dependent RNA polymerase (RdRp), which is expressed in a fused CP-RdRp form by ribosomal frameshifting [15,16]. During capsid assembly, one or a few CP-RdRp(s) are incorporated inside the virus particle [34,35]. The internal RdRps function for intraparticle genome synthesis is to synthesize nascent (+)ssRNAs that can exit via the pore(s) of the virus capsid [25,36,37]. Intraparticle genome synthesis and the RdRp/pore are considered fundamental requirements of non-enveloped icosahedral dsRNA viruses for sequestering the host's antiviral defense system, which could be triggered by viral dsRNAs [25,37]. However, the pores of some icosahedral dsRNA viruses that infect multicellular hosts (e.g., toti-like viruses) are obstructed [23,25,38]. Considering the unique phylogenetic clade of GLV in evolution, the surface and pore structures of GLV are key to deeply elucidating the infection and intraparticle genome synthesis mechanisms in the transmission steps of *Totiviridae* and toti-like viruses.

Recently, two subtypes of GLV, namely, GLV-HP and GLV-CAT, have been described in depth [15]. These subtypes have been found to infect the *G. duodenalis* Assemblage A isolate HP-1, of human origin, and CAT1, of cat origin. Despite limited nucleotide and amino acid divergence, GLV-HP and GLV-CAT show different phenotypes when individually infecting the naive *G. duodenalis* Assemblage A isolate WBC6 [15]. GLV-HP tends to form aggregates and accumulate below the trophozoite plasma membrane and inside the cell body, whereas GLV-CAT does not form aggregates, is scattered in the trophozoite, and does not accumulate in the cell body [15]. Additionally, chronic infection of GLV-HP hampers parasite growth more markedly than GLV-CAT, indicating that GLV-HP might have features of a pre-lytic virus [15].

The atomic models of the yeasts ScV-L-A and L-BC [35,39], protozoan TVV2 [40], and toti-like virus OmRV [24,25] have recently been determined; however, the structure of GLV is still limited to an intermediate resolution of 6 Å [20]. Here, we describe the first high-resolution structure and atomic model of GLV-HP and GLV-CAT virions, demonstrating the conserved and unique structural features of infection and intracellular genome synthesis in GLV and other *Totiviridae* or toti-like viruses. The comparison between GLV-HP and GLV-CAT also presents potential structural motifs that could greatly modify the transmission mechanisms and virulence of the GLV strains.

## Results and discussion

### Cryo-EM structural determination, capsid geometry, and atomic models

The first high-resolution capsid structure of two GLV subtypes was determined using cryo-genic electron microscopy (cryo-EM) single-particle analysis at a resolution of 2.1 Å and 2.6 Å for GLV-HP and GLV-CAT, respectively (Figs 1A, 1H, and S1 and S1 Table). As confirmed in our observations (Figs 1B, 1I, and S2) [15], the GLV-HP particles tend to aggregate with each other both within giardia parasite cells and in purified samples, whereas the GLV-CAT particles do not exhibit this behavior. The resolutions achieved from the obtained cryo-EM maps are the highest in the *Totiviridae* virus database to date, which enables us to discuss the capsid structure and CP–CP interactions more precisely.

The GLV capsid comprises 60 copies of CP dimers named subunits A and B (CP-A and CP-B) in the triangulation number (T) = 1 icosahedral asymmetric unit (Fig 1A, 1C, 1H and 1J). The translation of CP does not start with an AUG codon (S3 Fig) [15,33]. The atomic models of CP-A and CP-B, both in GLV-HP and GLV-CAT, were built from Ile70 or Lys71 residue into Asp928 or Val929 residue, apart from the CP-A of GLV-HP (Figs 1D–1G, 1K–1N, and S3). The CP's N-terminal residues (Residues 1–69) seem to be located on the interior side of the capsid and are not found in the cryo-EM maps (S3 Fig). The overall structure of the GLV CP shows a typical α-helix-rich α+β fold commonly adopted in the *Totiviridae* and other icosahedral dsRNA viruses (Figs 1C, 1J, and S3). The structures of CP-A and CP-B are very

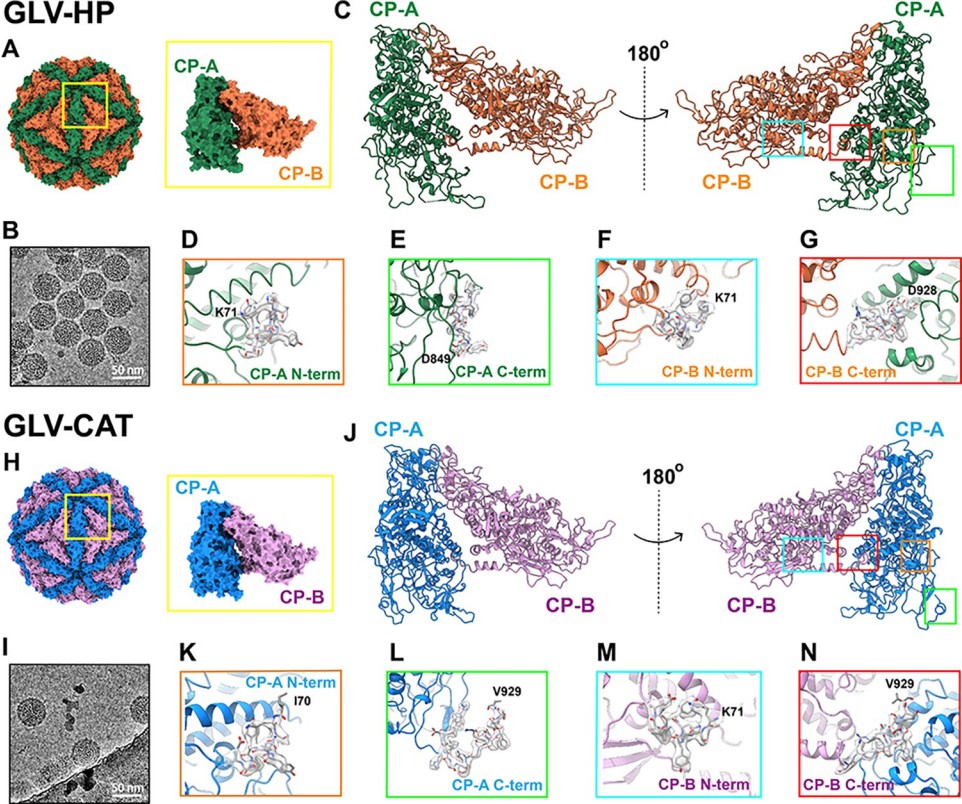

**Fig 1. Atomic model of capsid and CPs in GLV-HP and GLV-CAT.** The T = 1 capsid geometry and CP-A and CP-B organization in (A) GLV-HP and (H) GLV-CAT. Raw cryo-EM micrographs of (B) GLV-HP and (I) GLV-CAT. The close-up views of N-termini and C-termini in CP-A and CP-B are shown in (C–G) for GLV-HP and (J–N) for GLV-CAT.

similar, as the total root mean square deviation (RMSD) is 0.618 Å over the pruned set of Cα pairs both in the GLV-HP and GLV-CAT capsid (S4 Fig). However, the surface loops, termini (red dotted circles in S4 Fig), and interfaces between the two adjacent subunits contribute to overall higher RMSD values between CP-A and CP-B in the GLV-HP (4.338 Å across all 775 Cα pairs) and GLV-CAT (8.863 Å across all 848 Cα pairs) capsid.

## C-terminal extension and CP–CP interactions

In the GLV capsid structure, the CP–CP interactions are mainly mediated by the adjacent interfaces of CP-A and CP-B (Fig 1) and the short C-terminal extension that interlocks CP-A and CP-B on the interior side (Fig 2). The C-terminus of CP orients differently in CP-A and CP-B due to the interlocking between two adjacent subunits. Hence, this C-terminal interlocking is identified in two directions in the GLV-CAT capsid: CP-A to CP-B and CP-B to CP-A (Fig 2C–2E). However, only the CP-B to CP-A direction is identified in the GLV-HP capsid (Fig 2A and 2B). This C-terminal interlocking includes three possible hydrogen bonds: Arg671–Asp922, Arg675–Asp928, and Lys774–Asp927 in CP-B to CP-A in GLV-HP (Fig 2B). The C-terminal interlocking includes the same amino acid pairs as Arg671–Asp922 and Lys774–Asp927; however, more intersubunit interactions are identified in GLV-CAT (Fig 2E). In the CP-A to CP-B direction, only two possible hydrogen bonds—Arg675–Asp928 and E872–N917—are present in GLV-CAT (Fig 2D). In the ScV-L-BC capsid, only a very short C-terminal extension of the CP-A has been identified [35], and ScV-L-A, LRV-1, and TVV2 have no interlocking C-terminal extension [39–41]. Considering the C-terminal structure of these viruses, the C-terminal extension of GLV is longer and interlocks adjacent A and B subunits in both directions, at least in GLV-CAT (Fig 2C–2E). This interlocking mechanism in the GLV capsid is similar to that of the toti-like virus OmRV. The C-terminal extension of the OmRV-AK4 strain swaps the orientation on the Ser1603 residue in the A and B subunits to enable interlocking in both the A to B subunits and the B to A subunits [25]. In the OmRV-LZ strain, the long C-terminal extension of the B subunit interacts with several neighboring subunits [24]. In the C-terminal extension of the GLV-CAT subunits, similar swapping occurs structurally on the Lys902 residue. Other segmented and large icosahedral dsRNA viruses, such as the megabirnavirus and quadrivirus, show an extra-long C-terminal extension to interact with adjacent subunits or several proximal subunits [23,42].

It is interesting why the C-terminal extension is required for GLV, OmRV, and other segmented icosahedral dsRNA viruses. It has been previously hypothesized that the C-terminal extension may be required for the stability of the larger capsid as a consequence of packaging its own longer genome to tolerate the necessary higher interior pressure [23,25,35]. The capsid of ScV-L-A, ScV-L-BC, LRV-1, and TVV2 is 35–42 nm in diameter and packages a 4.5–5.5 kbp dsRNA [35,39–41], while that of the toti-like virus OmRV is 42 nm in diameter and packages a 7.5 kbp dsRNA [24,25]. The GLV capsid is 45 nm in diameter and packages a 6.2 kbp dsRNA [15], which implies that the interior pressure of the GLV capsid is slightly less than that of the OmRV capsid but more than or similar to those of the ScV-L-A, ScV-L-BC, and TVV2 capsids. Therefore, the length of the GLV C-terminal extension and its contribution to interlocking correspond to the expected interior pressure in *Totiviridae* and toti-like viruses. The C-terminal extension is also important for the interior localization of RdRp. However, the structure of the fused CP-RdRp has not been determined in the obtained cryo-EM map.

## Structural comparison of CPs within GLV, *Totiviridae* viruses, and other icosahedral dsRNA viruses

The CPs of T = 1 icosahedral dsRNA viruses share a conserved α-helix-rich α+β structural fold, as does GLV CPs (Fig 1); thus, their structural alignments can inform both unique and

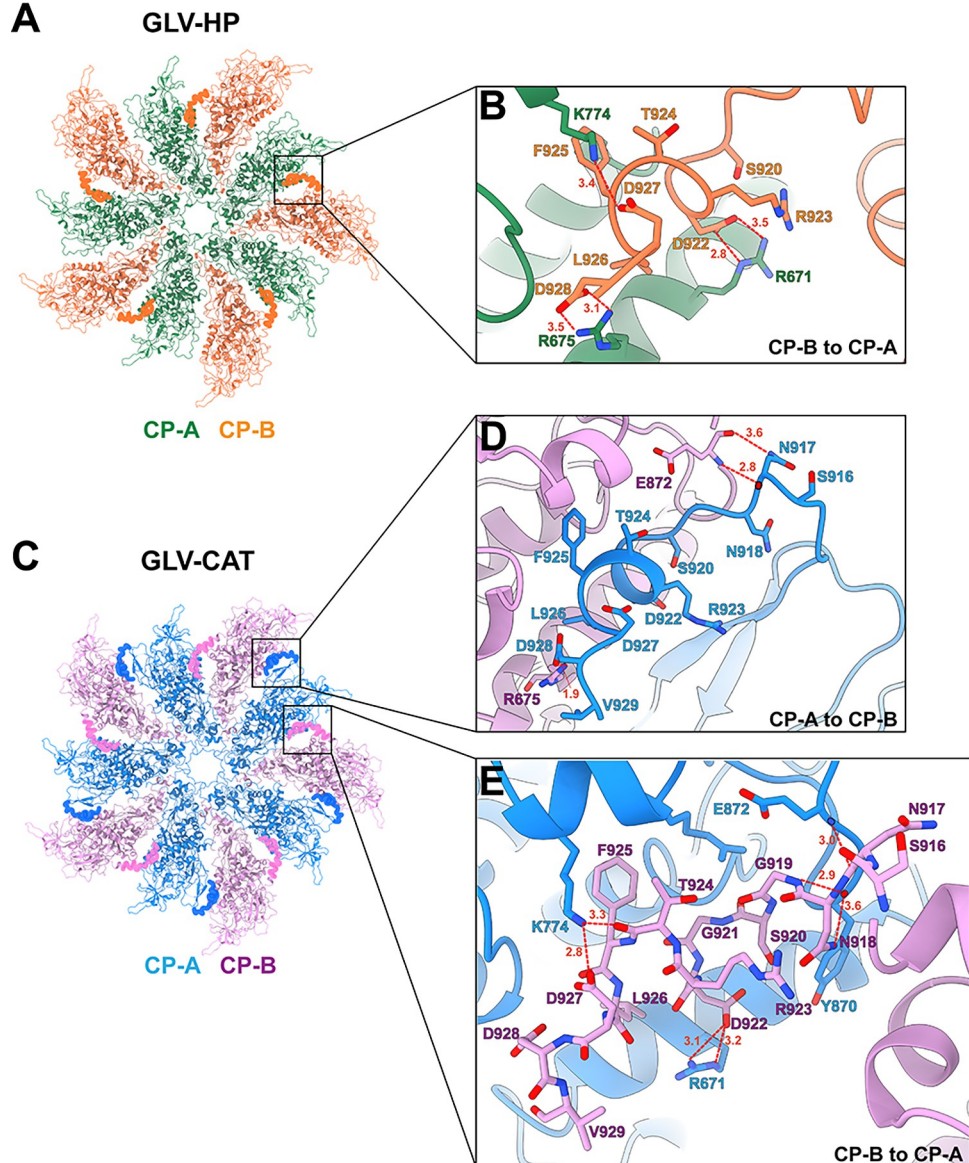

**Fig 2. C-terminal extension structure of the CP–CP interface in the GLV-HP and GLV-CAT capsids.** The atomic model of the 5-fold CP complex of (A) GLV-HP (CP-A in green; CP-B in orange) and (C) GLV-CAT (CP-A in light blue; CP-B in pink) is shown from their inside view. The interlocking C-terminal extensions from CP-A to CP-B and those from CP-B to CP-A are highlighted in the surface representation. The close-up views of the interactions between the C-terminal extension and the adjacent capsid are shown in (B) for GLV-HP and in (D) and (E) for GLV-CAT. The amino acid residues involved in the interactions are indicated. The red dashed lines between the amino acid residues indicate predicted hydrogen bonds with their distances in Ångström.

common structural features within these viruses. To search for CP structures that are akin to that of GLV in the latest PDB repository, the atomic model of the GLV CP was queried in the Dali online server, yielding eight similar CP structures of dsRNA viruses belonging to *Totiviridae*, including the unclassified toti-like virus OmRV, *Chrysoviridae*, *Megabirnaviridae*, and *Quardriviridae* viruses (S5A Fig). All the obtained atomic models were further aligned using MUSTANG and generated an RMSD-based structure phylogeny of these CPs (S5B Fig). Both results indicate that the closest CP structure of GLV is that of the toti-like virus OmRV and

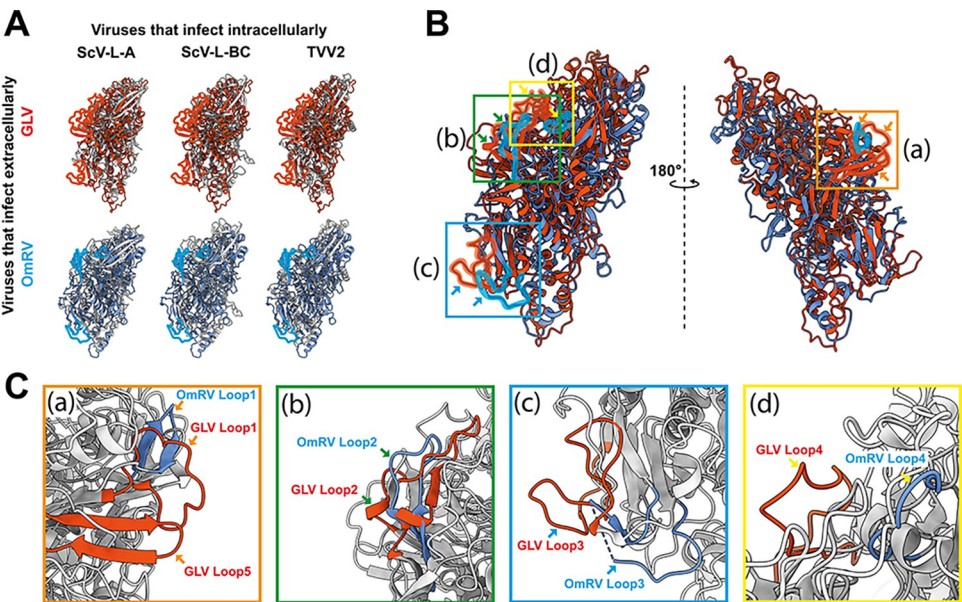

**Fig 3. Structural comparison of the GLV CP with those of *Totiviridae* and toti-like viruses.** ScV-L-A, ScV-L-BC, and TVV2 CPs are shown in grey. The GLV and OmRV CPs are shown in orange and blue. The unique surface loops of the GLV and the OmRV CPs are shown as transparent surface representation. (A) The CP of GLV and OmRV that infects extracellularly is aligned with that of ScV-L-A, ScV-L-BC, and TVV2, which infect intracellularly. (B) Aligned CP structure of GLV and OmRV. The colored boxes, also shown as (a), (b), (c), and (d), indicate the surface regions of the extra loops that are identified in the GLV and OmRV CPs. (C) Close-up views of the extra loops. The box colors and regions correspond to those in (B). CP-B subunit of GLV-HP (PDB ID: 8R0F), OmRV-LZ (PDB ID: 7D0K), ScV-L-A (PDB ID: 1M1C), ScV-L-BC (PDB ID: 7QWX), and TVV2 (PDB ID: 7LWY) were utilized for the structural alignment.

that it is distantly related to those of the other *Totiviridae* viruses infecting yeast and protozoa (S5 Fig). Total RMSD value between GLV and OmRV CP-B is 5.8 Å (Amino acid sequence coverage: 82%, TM-score = 0.65). Genetic phylogenetic analysis has demonstrated that GLV has its own cluster, which is located between *Totiviridae* and toti-like viruses [16,20]. The structural comparison between the CPs of GLV and OmRV CPs provides new insight into the unclarified relationship between *Totiviridae* and toti-like viruses, which might reflect their acquisition and loss of functional structures on the capsid.

Proceeding with this assumption, we compared the CP structure of yeast and protozoan *Totiviridae* viruses with the toti-like virus OmRV. The GLV CP presents two extra surface regions on the capsid surface compared with those of yeasts ScV-L-A and ScV-L-BC and protozoan TVV2, which infect intracellularly (Fig 3A), and the OmRV CP presents similar extra surface regions on the capsid (Fig 3A). These extra surface regions on the OmRV CP have previously been suggested as being unique to the toti-like virus CPs and not exhibited in the yeast ScV-L-A CP [25]. These surface regions are composed of extra loops and are located in amino acid residues 156–169 (loop 1), 364–388 (loop 2), 415–443 (loop 3), 527–546 (loop 4), and 565–583 (loop 5) in the GLV CP, and in amino acid residues 874–883 (loop 1), 1076–1095 (loop 2), 1115–1141 (loop 3), and 1541–1551 (loop 4) in the OmRV CP (boxed regions in Figs 3B and S6). One extra surface region is built up with loop 3 in the GLV and OmRV CPs (blue box or (c) in Fig 3B and 3C), whereas the other extra surface region is built with loops 1, 2, 4, and 5 in the GLV CP or loops 1, 2, and 4 in the OmRV CP (orange, green, and yellow boxes or (a), (b) and (d) in Fig 3B and 3C). The secondary-structure-based amino acid sequence alignment presents no obvious sequence identity between the OmRV loops and GLV loops,

although loops 1, 2, and 3 are located in topologically and conformationally similar regions (Figs 3C and S6). GLV and OmRV employ an extracellular phase, unlike other *Totiviridae* viruses, which only have an intracellular transmission mechanism [15,16,27].

The extracellular transmission mechanism of *Totiviridae* and toti-like viruses is still unclear; however, these surface loops could have profound roles in extracellular cell-to-cell transmission, such as particle release, membrane penetration, and receptor-binding mediated cell entry. The toti-like virus OmRV and IMNV and the megabirnavirus RnMBV1 also possess additional CrP on their surface, which may facilitate extracellular or horizontal cell-to-cell transmission in multicellular hosts [23,24,43]. Considering that the GLV capsid does not express and has no extra surface CrP structurally (Fig 1), CrP must be a non-essential factor in cell-to-cell transmission. These findings show agreement with previous results that indicate that inhibiting CrP interaction with the OmRV capsid does not eliminate the infectivity of OmRV [24,28]. Since the amino acid sequence of CrP is not predicted to be a cell-penetrating peptide, CrP might only be implicated in receptor-binding and subsequent endocytosis. Further, OmRV particles without CrPs are still infectious to host mosquito cells [34]. To fully understand the transmission mechanisms of *Totiviridae* and toti-like viruses, it is necessary to study these identified extra surface loops.

## 5-fold pore structure

The *Totiviridae* viruses have a pore on each 5-fold axis [35,40] or each 2-fold/3-fold axis [44], possibly for importing cellular nucleotide triphosphates (NTPs) to the intraparticly packaged RdRp(s) and for exporting the synthesized viral (+)ssRNA to the cytosol. In *Reoviridae* viruses, the pore on the T = 1 inner capsid dynamically and structurally synchronizes with transcribing the nascent viral (+)ssRNAs on the interior RdRp in situ [36,45–47]. The structural mechanisms of the pores in *Totiviridae* viruses are seldom described due to a lack of in-situ transcribing structures. Asymmetric reconstructions were attempted to determine the structure of the CP-RdRp; however, its localization within the capsid remained ambiguous (S7 Fig). Only a few or one RdRp are randomly incorporated into their capsid particles, unlike *Reoviridae* viruses [35], although a partially spooled genome structure like that of *Reoviridae* viruses is observed in capsid-subtracted 2D class projections in our GLV structure (S8 Fig).

GLV-HP and GLV-CAT capsid structures exhibit pores on each 5-fold axis with an outer and inner diameter of 10–11 Å and 19–20 Å, respectively (Fig 4), similar to other *Totiviridae* viruses [25,35,40]. At the center of the pore, the surface is positively charged because of the cluster of the 10 lysine residues (five Lys209 and five Lys217 residues) (the outside view is shown in Fig 4). In contrast, the interior side of the 5-fold pore is negatively charged (the inside view is shown in Fig 4). These positive and negative surface properties are constantly observed in other *Totiviridae* viruses [25,35,40], which implies their essential functions. The dsRNA viruses infecting multicellular hosts possess similar pores composed of two consecutive arginine residues (RR motif); however, the pores are obstructed [23,25,34,38]. In toti-like viruses and human picobirnavirus, simple conformational changes on the pore assist them in opening [25,38].

The electropositive capsid surface could have a role in recruiting negatively charged NTPs to the pores, as previously suggested for icosahedral dsRNA viruses [23,25]. A capsid of the human immunodeficiency virus (HIV) also has positively charged pores for recruiting deoxynucleotide triphosphates (dNTPs) for its intraparticle genome synthesis [48,49]. However, the positively charged residues of the HIV capsid pores are also critical for trapping and utilizing cellular inositol phosphates (IP5 or IP6) during capsid assembly and maturation [50,51]. These findings and previous results imply the multifunctionality and importance of the two Lys residues of the GLV pore.

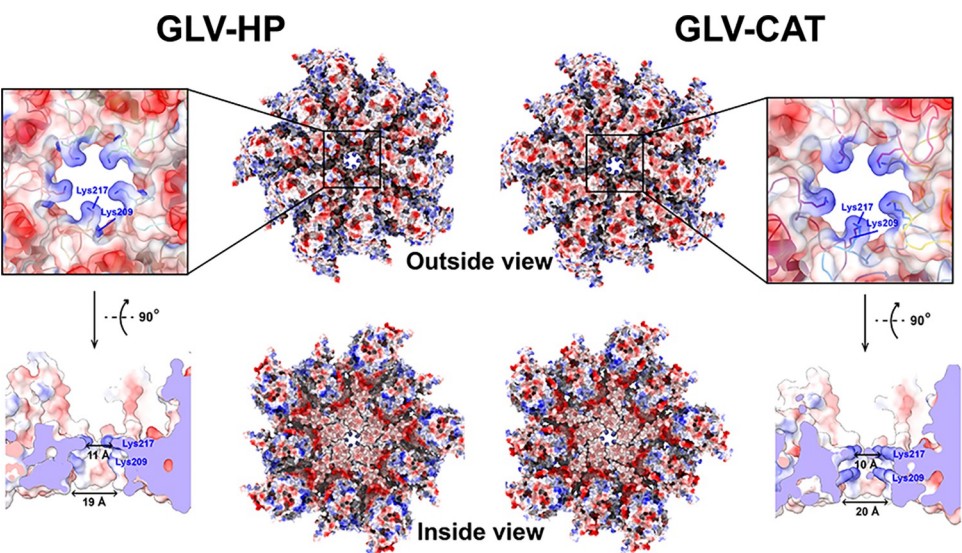

**Fig 4. Pore structure of the GLV capsid.** A surface electrostatic potential map of the 5-fold CP complex of GLV-HP and GLV-CAT from outside and inside views is shown in red (negatively charged) and blue (positively charged) scales. A close-up view and cross-section of the pore structure and surface charges are also shown at the bottom.

## Lack of putative cap-snatching active pocket

In yeast *Totiviridae* viruses ScV-L-A and ScV-L-BC, a cap-snatching pocket that exhibits the His residue, His154 in ScV-L-A or His156 in ScV-L-BC, is structurally located on similar regions of each respective capsid [35,39] (Fig 5A). Although the active pocket is built with a certain number of amino acid residues, only the invariant His is known to be functional [32]. It is speculated that the protozoan TVV2 CP could have its cap-snatching pocket in a structurally similar location to the yeast CPs [40] (Fig 5A); however, this is still under discussion. In the GLV and OmRV CPs, the cap-snatching pocket and the invariant His residue cannot be found in the position corresponding to that of the yeast ScV-L-A and ScV-L-BC CPs (Fig 5A). Structural alignments of *Totiviridae* and toti-like viruses revealed that His154 in the ScV-L-A and His156 in the ScV-L-BC CPs are located on the surface region, which is followed by structurally well-conserved helices in the *Totiviridae* and toti-like viruses' CPs (Fig 5B). Due to the swapped orientation of the secondary structural elements after the structurally conserved helices in TVV2, GLV, and OmRV CPs, their corresponding His residue is not located structurally in the previously reported putative cap-snatching pocket (Fig 5B). Other viral methyltransferases, such as flavivirus NS5 protein, and cap-snatching complexes, such as influenza virus PA/PB1/PB2 complex, have a cavity formed by aromatic amino acid residues (Trp, Tyr, and Phe). These viral enzymes show a negatively charged active site next to the positively charged cluster [29,30,52]. In the atomic model of GLV and OmRV CPs, no such charged surface is evident in the surface area of the His residues (His288, His387, and His444 in GLV or His986 in OmRV) (Fig 5A). Therefore, these observations strongly suggest that the cap-snatching pocket no longer structurally exists in GLV and OmRV CPs and that the capsids of the *Totiviridae* viruses have lost the cap-snatching function through evolution. This finding is consistent with the lack of a 5'-cap RNA structure in GLV transcripts [15,33].

## Structural differences between GLV-HP and GLV-CAT

GLV-HP and GLV-CAT present distinct intracellular localization and extracellular release efficiency when chronically infecting the *G. duodenalis* isolate WBC6, which could, in turn, reflect

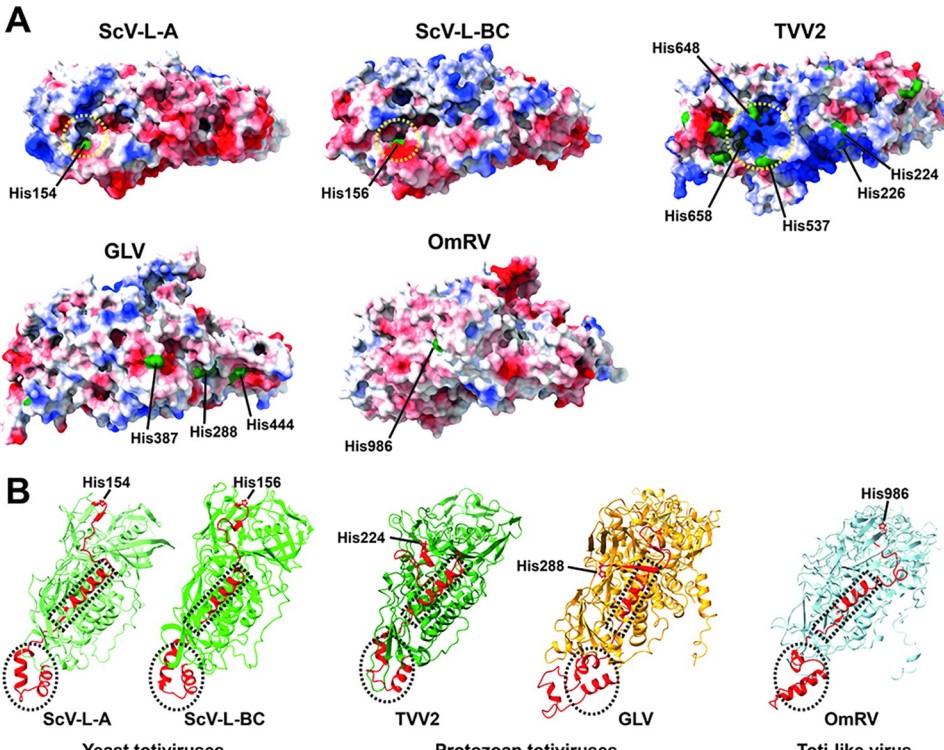

**Fig 5. Lack of putative cap-snatching pockets in GLV and OmRV. (A)** Electrostatic surfaces of *Totiviridae* and toti-like virus CPs are shown in blue (positive) and red (negative) scales. The CP of the yeast *Totiviridae* viruses ScV-L-A and ScV-L-BC have cap-snatching active pockets and invariant His residues (His154 or His156) (yellow dotted circles). The CP of TVV2 is speculated to have a cap-snatching pocket with three putative His residues (His537, His648, and His658) in a position similar to that in the yeast *Totiviridae* viruses (yellow dotted circle). However, in the GLV and OmRV CPs, no His residue is observed on the corresponding cap-snatching pocket, although three His residues (His288, His387, His444) in the GLV CP and one His residue (His986) in the OmRV CP are exhibited on the surfaces. **(B)** Conserved helices and invariant His residue in *Totiviridae* and toti-like viruses. A conserved helix-turn-helix and a long helix are highlighted as dotted circles and rectangles in each CP structure.

their virulence [15]. The GLV-HP particles interact with each other (Figs 1B, 1I, and S2), which causes particle aggregation and affects the preferable intracellular localization [15]. Such particle-to-particle interactions should be mediated by the surface of the GLV-HP capsid. Therefore, the structural differences between GLV-HP and GLV-CAT CPs are the focus of much research.

Within the GLV-HP and GLV-CAT CP amino acid sequences, 45 amino acid alterations exist in the structurally visible region (Ile70–Val929) (Figs 6A and S9). Although the capsid surface of GLV-HP seems less charged (Fig 4), these amino acid alterations do not dramatically alter the hydrophobicity and charges on the capsid surface. However, apparent structural differences are observed between GLV-HP and GLV-CAT CP-A/CP-B dimers (Figs 6B and S10). The major structural difference of CP-A is the aforementioned absence of C-terminal extension in GLV-HP (Figs 2 and S10), while that of CP-B is a variable loop that consists of Pro316–Ser330 residues, localizing between the interface of CP-B and that of CP-A (Figs 6B and S10). Both CP-A and CP-B show high RMSD in the proximal to the 5-fold axis of the capsid (Figs 6B and S10).

Notably, Phe320 in GLV-HP forms intrasubunit interactions with Val184, Leu590, and Asn592 in the CP-B (Fig 6C); however, in GLV-CAT, the corresponding Leu320 in CP-B interacts with Asn112 and Gln282 residues in the adjacent CP-A (Fig 6D). The switching

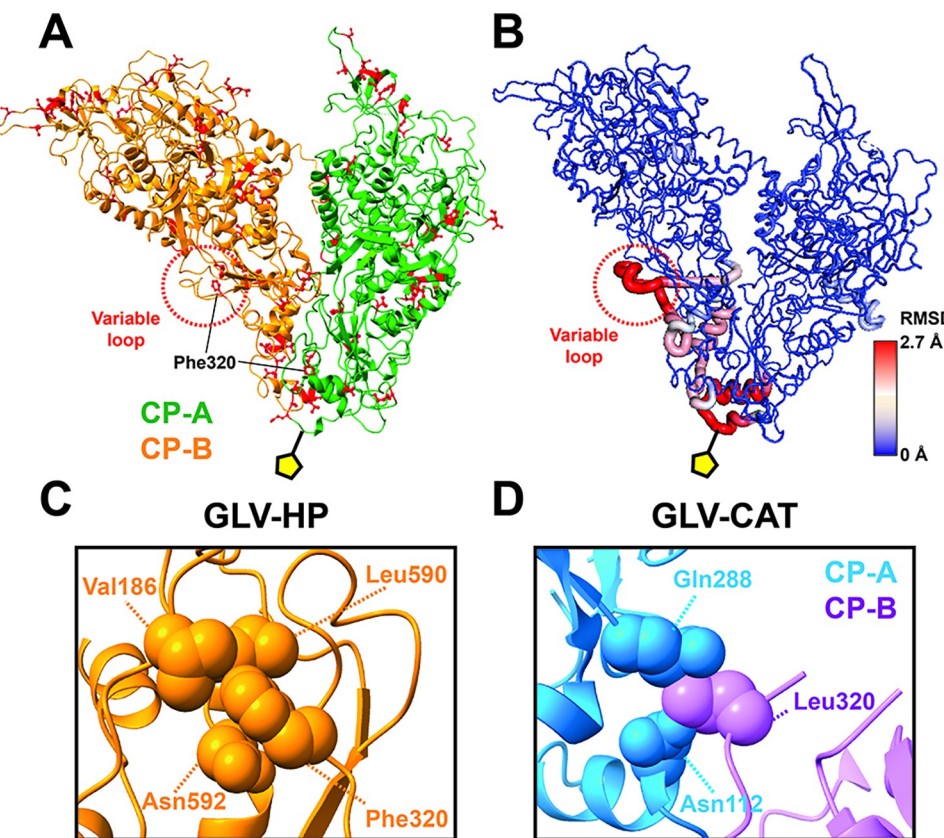

**Fig 6. Amino acid alterations and structural variability of CPs between GLV-HP and GLV-CAT.** (A) Amino acid alterations between GLV-HP and GLV-CAT are mapped on a GLV-HP CP-A/CP-B dimer. (B) Local RMSD values are calculated per amino acid residue. The RMSD values are scaled by blue (low RMSD) and red (high RMSD). The yellow pentagon indicates the 5-fold axis of the GLV capsid. (C) Intra-subunit interaction of Phe230 of CP-B in the GLV-HP capsid. (D) Inter-subunit interaction between CP-A and Leu320 of CP-B in the GLV-CAT capsid.

between the intrasubunit and intersubunit interactions by the F320L point mutation largely contributes to driving the observed structural change in the variable loop. The Phe320 or Leu320 residue in the CP-A is located near the 5-fold axis, which might also affect the observed structural variation there (Fig 6A and 6B). This additional CP–CP interaction, which is mediated by Leu320 and the variable loop in GLV-CAT, should make the CPs' network in the capsid more robust. With the more robust C-terminal interlocking in the GLV-CAT capsid (Fig 2), the capsid assembly of GLV-CAT is structurally distinct from that of GLV-HP. Nano differential scanning fluorimetry (nanoDSF) thermographs differences in Ti (inflection temperature) values between GLV-HP and GLV-CAT particles, which may reflect their capsid stability in multiple dissociation or unfolding steps (S11 Fig). It is also another possibility that a lack of intersubunit interactions, which are mediated by the variable loops within the GLV-HP capsid, might instead allow intersubunit interactions between two virions.

## Conclusion

The comparative analysis of the first high-resolution atomic model of two GLV prototypes enables us to present structural insights into acquired and lost functional features in the evolution of dsRNA *Totiviridae* and toti-like viruses. The two Lys residues that compose the GLV pores are critical for understanding intraparticle genome synthesis and particle assembly in

GLV and other *Totiviridae* viruses. The surface loops that only exhibit in the OmRV and the GLV capsid explain an essential requirement for acquiring extracellular cell-to-cell transmission, in particular, virus entry and egress. The comparative analysis of the two GLV prototypes reveals significant conformational differences in the C-terminal extension and the variable loop, which are critical for CP–CP interactions. It is hypothesized that the loose assembly of the CPs in GLV-HP could expose hydrophobic CP–CP interfaces to facilitate particle–particle interactions. This hypothesis is in agreement with observations in other viral models, such as the VLPs formed by VP40, the MCP of (-)ssRNA Ebola virus, for which mutations at a specific loop in the protein N-terminus alter both protein oligomerization and virus egress [53]. Interestingly, these structural changes are potentially responsible for the transmission and virulence of the two GLV strains, which will provide strategies to engineer GLV to enhance its pathogenic effects on *G. duodenalis*.

## Materials and methods

### Parasite culture and viral particle preparation

The *G. duodenalis* isolates used in this study comprised HP-1 (Assemblage AI) originally isolated from a human patient [54], and a kind gift from Prof. E. Nohynková (Charles University of Prague, Czech Republic), and CAT-1 (Assemblage AI) originally isolated from a cat [55], and a kind gift from Dr. G.S. Visvesvara (Centers for Disease Control and Prevention, Atlanta, Georgia, USA). HP-1 and CAT-1 are chronically infected with the virus strains GLV-HP and GLV-CAT, respectively [15]. Trophozoites were routinely grown in axenic, microaerophilic conditions in a TYI-S33 medium, supplemented with 10% (v/v) adult bovine serum (Euroclone S.p.A., Milan, Italy) and 0.05% (v/v) bovine bile (Sigma-Aldrich, Merck Life Science S.r. l., Milan, Italy) at 37˚C in 10 mL screw-cap tubes (Nunc, Thermo Fisher Scientific, Waltham, MA, USA) and sub-cultured when confluence was reached (every 48–72 h). For viral particle purification, 1 L of the trophozoite culture of each isolate was grown in 50 mL screw-cap tubes (Falcon, Thermo Fisher Scientific, Waltham, MA, USA) with a TYI-S33 medium for 60 h. The tubes were chilled on ice to detach trophozoites, and the parasites were harvested by centrifugation at $900 \times g$ for 10 min at 4˚C. Trophozoite pellets were combined and washed twice with cold PBS by centrifugation at $900 \times g$ for 10 min at 4˚C. Trophozoite pellets were resuspended in 8 mL of cold PBS and lysed by sonication (5–6 times for 30 s at 60% power and a 10% duty cycle) with a Sonoplus ultrasonic homogenizer (Bandelin Electronic, Berlin, Germany). The lysate was centrifuged at $10,400 \times g$ (in JA20 centrifuge, Beckman) for 10 min at 4˚C to remove debris. To sediment the viral particles, the supernatant was layered onto a PBS/1.5 M sucrose solution at a sample/sucrose 4:1 ratio and ultracentrifuged at $264,000 \times g$ on an Optima TLA 100.3 ultracentrifuge (Beckman) for 2 h at 4˚C. The virus pellet was resuspended with 8 mL of cold PBS, CsCl was added at a final density of 1.39 g/mL, and the volume was adjusted to 12 mL with a PBS/CsCl solution at the same density. Virions were banded by density gradient centrifugation at $152,000 \times g$ for 16 h at 4˚C in a SW41 rotor (Beckman Coulter SRL, Milan, Italy). The gradient was then fractionated, and virion-positive fractions were identified by the phenol extraction of nucleic acids, as previously described [15]. The positive fractions were pooled and dialyzed overnight in sterile PBS/glycerol 20% (v/v) and stored at -80˚C until use. The overall quality of the prepared purified virions was examined by transmission electron microscopy (TEM), as previously described [15]. For the cryo-EM grid preparation, the purified GLV-HP and GLV-CAT particles were further concentrated to 20–30 μL (approximately $10^9$–$10^{10}$ particles/mL).

## Cryo-EM data acquisition and map reconstruction

Three microliters of the purified GLV-HP or GLV-CAT sample was placed on a glow-discharged holey carbon grid (Quantifoil R2/2 or R2/1, Cu 300 mesh; Quantifoil Micro Tools GmbH) for 3 s of blotting time at 4˚C and 100% humidity with filter paper and then plunge-frozen in liquid ethane using the Vitrobot Mark IV (Thermo Fisher Scientific). The prepared grids were pre-screened with a 200-kV Glacios cryo-EM at the Uppsala Cryo-EM center. The full datasets of GLV-HP and GLV-CAT were collected with a 300-kV Titan Krios G2 cryo-EM (Thermo Fisher Scientific) equipped with a Gatan K3 BioQuantum detector and the post-column electron energy filter (20 eV slit width) at the SciLifeLab Cryo-EM Infrastructure Unit. The image movies were collected in a counted super-resolution mode at a nominal magnification of 81,000×, which corresponds to a pixel size of 1.06 Å /pixel. The defocus range is 0.2 μm steps in the range of 0.7–1.5 μm under focus. The total exposure per movie was adjusted to 30 $e^{-}/Å^{2}$ for 1.9 s and dose-fractionated into 30 frames. A total of 19,580 and 7,098 movies were collected for further image analysis. The parameters of the cryo-EM data collection are shown in S1 Table. The three-dimensional (3D) cryo-EM map of the capsid was reconstructed using CryoSPARC version 4.3.1 [56] linked to a local GPU/CPU computer cluster. The image frames were corrected by the patch motion correction using frames 3–28, and then the contrast transfer function (CTF) parameters were estimated using the patch CTF. The first template images of the virus particles were generated from 100–200 manually picked particles, and the templates were then used to automatically pick up virus particles using the template picker. Both the picked GLV-HP and GLV-CAT images contain approximately 20% empty particles; otherwise, all of them are filled particles. After a couple of 2D classifications, good 2D classes containing 28,342 and 5,804 good particles of GLV-HP and GLV-CAT, respectively, were selected. The final 3D map reconstruction was accomplished using the Homogeneous Refinement option by imposing icosahedral symmetry and an initial model that was generated by 20-Å lowpass filtering to a previously determined OmRV cryo-EM map [25]. Throughout the reconstruction process, the per-particle defocus, CTF parameters, spherical aberration, beam tetrafoil, and beam anisotropic magnification were optimized to enhance the map qualities. The final 3D maps of GLV-HP and GLV-CAT were determined to have a resolution of 2.1 Å and 2.6 Å, respectively, using the gold standard Fourier shell correlation (FSC) at a 0.143 cutoff (S1 Fig). These maps were subsequently employed to construct atomic models of the CPs.

## Atomic modeling and refinement

The first atomic models of GLV CPs (A and B subunits) were manually built in the cryo-EM map of GLV-HP using Coot version 1.0.06 [57]. The manually built atomic model was further refined iteratively using PHENIX 1.20.1 [58] and Coot. The amino acid residues of the refined atomic model of GLV-HP CPs were then mutated to those of GLV-CAT and placed on a GLV-CAT cryo-EM map. The GLV-CAT CPs were then refined using PHENIX and COOT, as described above. The validation statistics of the atomic models and the cross-correlation to the cryo-EM maps are shown in S1 Table.

## Structural analysis

To render the cryo-EM maps and atomic models, the UCSF Chimera and ChimeraX were used [59,60]. The viral CP structures resembling that of GLV-HP were comprehensively searched against all PDB-deposited structures using the Dali structure comparison server [61]. Eight CP structures in non-enveloped icosahedral dsRNA viruses were detected as similar to the CP of GLV-HP. These eight CP structures and the GLV-HP structure were superimposed by pairwise structure-based alignments using the MUSTANG program [62]. During the

alignments, all-to-all RMSD values were calculated in the superimposed structures. These RMSD values served as a distant matrix to generate a structural phylogeny of the viral CPs using a neighbor-joining method in MEGA X [63], as already described earlier [64,65]. The aligned structures were also visualized to discover conserved, acquired, and/or lost unique structures in the GLV CP. To calculate the local RMSD, the local_rms command was applied for the aligned GLV-HP and GLV-CAT CP dimers using the PyMOL script collection (PSICO) module [66].

## Supporting information

**S1 Fig. Overall and local resolution estimations of GLV-HP and GLV-CAT reconstructions.** (A) FSC curves and (B) local resolution of the final cryo-EM 3D reconstruction for GLV-HP and GLV-CAT.
(TIF)

**S2 Fig. Representative raw micrographs of GLV-HP and GLV-CAT particles.** Yellow arrows indicate empty particles.
(TIF)

**S3 Fig. Assigned amino acid residues and the secondary structure diagram of the atomic model of CP-B of GLV-HP and GLV-CAT.** The rainbow color begins with blue at the N-terminus (Pro1) to red at the C-terminus (Val929). Considering the predicted internal IRES sequence in the GLV genome, the translation of ORF1 (CP) does not initiate from the first methionine but from an internal amino acid residue. The first amino acid residue of the CP was started from the internal Pro residue (PENIT . . .), according to a previous mass spectrometry analysis of the purified GLV particles.
(TIF)

**S4 Fig. Structural comparison of CP-A and CP-B.** The CP-A and CP-B of GLV-HP are colored in green and orange, and those of GLV-CAT are colored in light blue and purple. The total RMSD values between CP-A and CP-B were calculated for GLV-HP and GLV-CAT. Some conformational changes are observed in the regions indicated by red dotted circles. The major core domain apart from N- and C-termini, and red dotted circled regions, are well aligned with RMSD = 0.618 Å over 684 Cα pairs between GLV-HP CP-A and CP-B), and 0.618 Å over 732 Cα pairs between GLV-CAT CP-A and CP-B while other loops and interfaces contribute to the overall higher RMSD values (4.338 Å across all 775 Cα pairs in GLV-HP, and 8.863 Å across all 848 Cα pairs in GLV-CAT).
(TIF)

**S5 Fig. DALI search and RMSD-based structural phylogeny of CPs in non-enveloped icosahedral dsRNA viruses. (A)** List of the eight identified CPs that are similar to the GLV CP obtained from a Dali search (Z score: 3.4–10.4). (B) RMSD-based structural phylogeny generated by the alignment of the GLV CP with the eight CP structures in a Dali search.
(TIF)

**S6 Fig. Secondary structure-based amino acid sequence alignment.** Amino acid sequences of GLV-HP (PDB ID: 8R0F, chain A), OmRV-LZ (PDB ID: 7D0K, chain A), ScV-L-A (PDB ID: 1M1C, chain A), ScV-L-BC (PDB ID: 7QWX, chain A), and TVV2 (PDB ID: 7LWY, chain A) CPs were aligned using PROMALS3D multiple sequence and structure alignment server. Extra surface loops of CP in GLV (loops 1–5) and OmRV (loops 1–4) are boxed in the aligned sequences.
(TIF)

**S7 Fig. Center slice of GLV-HP and GLV-CAT icosahedral and asymmetric reconstructions.** All images were generated from unmasked models. Asymmetric reconstructions were generated using symmetry expansion and local reconstruction options (Sym Exp) or the symmetry relaxation option (Sym Relax) in cryoSPARC software. A mask that covered one 5-fold vertex of the virus was utilized for the local reconstruction. The box size of the central slice is 640 x 640 pixels (1.06 Å/pixel) in the reconstruction with original particle images or 320 x 320 pixels (2.12 Å/pixel) in the reconstruction with downsampled particle images.
(TIF)

**S8 Fig. Selected 2D class-averaged projections of capsid-subtracted GLV-HP particle images.** The striped genome (side views, red squares as examples) and the spiral genome (top views, blue squares as examples) are observed, which are typical 2D classes of a partially spooled genome organization in *Reoviridae* viruses.
(TIF)

**S9 Fig. Amino acid sequence alignment of CPs between the GLV-HP and GLV-CAT strains.** The alignment was generated by CLC Sequence Viewer 7.0.
(TIF)

**S10 Fig. Structural differences between CP-A and CP-B in GLV-HP and GLV-CAT.** The overall RMSD values were calculated to evaluate the structural similarity. The major core domain are well aligned with RMSD = 0.401 Å across 773 Cα pairs between CP-As of GLV-HP and GLV-CAT, and 0.396 Å across 835 Cα pairs between CP-Bs of GLV-HP and GLV-CAT. Conformational changes were observed in the regions of C-terminal extension in CP-A and variable loop in CP-B, as indicated by yellow dotted circles.
(TIF)

**S11 Fig. NanoDSF thermographs of GLV-HP and GLV-CAT.** The thermographs and Ti values were obtained from purified samples of GLV-HP and GLV-CAT using Tycho NT.6 (NanoTemper).
(TIF)

**S1 Table. Cryo-EM data collection, refinement, and validation statistics.**
(TIFF)

## Acknowledgments

We acknowledge the use of the Cryo-EM Uppsala facility for sample vitrification and initial grid screening, funded by the Department of Cell and Molecular Biology, the Disciplinary Domains of Science and Technology and of Medicine and Pharmacy at Uppsala University. The data were collected at the Cryo-EM Swedish National Facility funded by the Knut and Alice Wallenberg, Erling Persson Family, and Kempe Foundations, and the SciLifeLab, Stockholm University and Umeå University. We thank Julian Conrad for help with data acquisition and Björn Persson for reconstructing the initial low-resolution cryo-EM maps of the GLV-HP and the GLV-CA. We also want to thank Prof. Eva Nohýnková, Charles University of Prague, Czechia, and Dr. Govinda S. Visvesvara, Centers for Disease Control and Prevention, Atlanta, Georgia, for their kind gifts of HP-1 and CAT-1 *G. duodenalis* isolates, respectively. A.M. acknowledges support from DESY (Hamburg, Germany), a member of the Helmholtz Association HGF.

## Author Contributions

**Conceptualization:** Marco Lalle, Kenta Okamoto.

**Data curation:** Han Wang, Gianluca Marucci, Anna Munke, Marco Lalle, Kenta Okamoto.

**Formal analysis:** Han Wang, Anna Munke, Mohammad Maruf Hassan, Marco Lalle, Kenta Okamoto.

**Funding acquisition:** Marco Lalle, Kenta Okamoto.

**Methodology:** Marco Lalle, Kenta Okamoto.

**Project administration:** Marco Lalle, Kenta Okamoto.

**Resources:** Gianluca Marucci, Marco Lalle.

**Supervision:** Marco Lalle, Kenta Okamoto.

**Validation:** Han Wang, Anna Munke, Mohammad Maruf Hassan, Marco Lalle, Kenta Okamoto.

**Visualization:** Han Wang, Kenta Okamoto.

**Writing – original draft:** Han Wang, Marco Lalle, Kenta Okamoto.

**Writing – review & editing:** Han Wang, Gianluca Marucci, Anna Munke, Marco Lalle, Kenta Okamoto.

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
