## [Decision Letter · Decision Letter 0]

4 Jan 2024

Dear Dr. Okamoto,

Thank you very much for submitting your manuscript "High-resolution comparative atomic structures of two Giardiavirus prototypes infecting G. duodenalisparasite" for consideration at PLOS Pathogens. As with all papers reviewed by the journal, your manuscript was reviewed by members of the editorial board and by two independent reviewers. The reviewers appreciated that the high-resolution structures of the totivirus particles presented are an important contribution, but they also raised a number on important issues that reauire your attention. In light of the reviews (below this email), we would like to invite the resubmission of a significantly-revised version that takes into account the reviewers' comments.

We cannot make any decision about publication until we have seen the revised manuscript and your response to the reviewers' comments. Your revised manuscript is also likely to be sent to reviewers for further evaluation.

Sincerely,

Félix A. Rey

Academic Editor

PLOS Pathogens

Guangxiang Luo

Section Editor

PLOS Pathogens

Kasturi Haldar

Editor-in-Chief

PLOS Pathogens

orcid.org/0000-0001-5065-158X

Michael Malim

Editor-in-Chief

PLOS Pathogens

orcid.org/0000-0002-7699-2064

Reviewer's Responses to Questions

**Part I - Summary**

Reviewer #1: Manuscript by Wang et. al describes the structures of two Giardia lamblia virus isolates belonging to the Totiviridae family. The two isolates GLV-HP and GLV-CAT have intracellular localization patterns. GLV-HP particles interact with each other and the possible reasons for the differences between the isolates is explored. The manuscript analyzes the differences between coat protein interactions at various interfaces and also compares the coat protein structures to other known Totiviridae virus structures. Overall, the structures are beautiful and the manuscript is well-written and easy to follow. I only have few major comments which will improve the overall impact of the manuscript.

Reviewer #2: In this manuscript by Wang et al, provide high-resolution cryo-EM structures of two giardia lamblia virus (GLV) strains, GLV-HP and GLV-CAT. These dsRNA totiviruses infect, through an extracellular step, the parasite giardia lamblia. As there is interest in using these viruses as potential “virotherapy”, the high-resolution structures provided in this MS provide structural details that may prove critical in this regard. Furthermore, they expand our structural understanding of the Totiviridae family allowing for evolutionary analyses. While the structures are thus informative, the MS could be improved to provide more thorough analyses of the structures as in parts it is rather superficial.

**Part II – Major Issues: Key Experiments Required for Acceptance**

Reviewer #1: 1. How do the unmasked structures look and how is the genome organized inside the capsid. Providing a central slice of unmasked maps in the suplementary material is valuable.

2. Did the authors observe any empty particles as observed in OmRV structures ?

2. The authors only imposed icosahedral symmetry on the particles. Have the authors tried symmetry expansion and localized reconstruction of the five-folds to see if they can visualize any more details beyond whats observed from the icosahedral reconstruction.

3. The newest version of cryosparc has a symmetry relaxation option and I can understand that the manuscript may have been submitted before the release. But it will be very useful to try this and see if we can see any features not visible in the icosahedral reconstruction. Especially for GLV-CAT because there are some features in the mask generated by cryosparc in FigS1.

4. Figure 3 is such an important and valuable figure but the color choices are not helpful to see the differences. This figure needs to be redesigned so that it can convey what the authors want the readers to understand. Please choose a color palette that is also friendly for color blindness.

5. Please provide raw micrographs of both GLV-HP and GLV-CAT in the supplementary section.

Reviewer #2: 1) The authors state that the GLV-CAT is “more stable” than GLV-HP (Ln 383) has this been experimentally determined or is this simply conjecture? If the latter, the authors should substantiate this claim by acquiring Tm of the different particles.

2) Is it possible to do mutagenesis (or perhaps it has already been done?) on the extra surface loops to validate their role in transmission? The authors should revise Fig 3 as it is rather difficult to interpret. More details should be provided regarding the amino acid identity within these loops. At present simply representing their secondary structure and amino acid numbers (lines 265-268) is too superficial.

**Part III – Minor Issues: Editorial and Data Presentation Modifications**

Reviewer #1: (No Response)

Reviewer #2: 1) This is more of a philosophical question but the authors use “acquired” to describe the extracellular lifecycle of GLV (ln268, and elsewhere). Is it truly that they acquired it or did the other totiviruses/toti-like viruses “lose” this capability?

2) In Figure 2B, D, E, distances should be reported.

3) The Discussion needs to be expanded to contextualize the authors findings. It may be appropriate to move lines 221 -236 into the Discussion. Indeed, in many of the sections contain prose that would be better suited for the Discussion.

4) The manuscript would benefit from a thorough editing for clarity as well as fixing grammatical and typos. In some cases, the word choice is inappropriate. Below are some examples but it is not an exhaustive list.

1. Ln 114: “methylguanocine” should be “methylguanosine”

2. Ln 147: a space is missing between duodenalis and Assemblage

3. Ln 149: The authors, I think mean “GLV-CAT does not form aggregates”.

4. Ln 159, ln 258 (and elsewhere): the use of “intensive” should be removed.

5. Ln 161 “lifestyle” is the wrong word choice.

6. Ln 178 “settled” is the wrong word choice.

7. Ln238 “relations” is the wrong word choice; perhaps “comparison”.

8. Ln293 “structurally collaborates” is wrong word choice

PLOS authors have the option to publish the peer review history of their article (what does this mean?). If published, this will include your full peer review and any attached files.

Reviewer #1: No

Reviewer #2: No
---

## [Editor Report · Decision Letter 1]

22 Feb 2024

Dear Dr. Okamoto,

Thank you very much for submitting your manuscript "High-resolution comparative atomic structures of two Giardiavirus prototypes infecting G. duodenalisparasite" for consideration at PLOS Pathogens. As with all papers reviewed by the journal, your manuscript was reviewed by members of the editorial board and by several independent reviewers. The reviewers appreciated the attention to an important topic. Based on the reviews, we are likely to accept this manuscript for publication, providing that you modify the manuscript according to the review recommendations.

The revised manuscript by Wang et al is improved with respect to the original version. The comparative study of the two GLV isolates is interesting, and the results are informative to better understand the life cycle of these dsRNA viruses. The manuscript, however, lacks clarity at certain places, some of which are specified below. Also, it is important that the manuscript be seen by a native English speaker, as the narrative is sometimes ambiguous and difficult to follow. I raise some of the issues below, as well as comments on the Figures.

Line 168-169: “the GLV-HP particles tend to interact with each other”: does this mean the purified particle preparation tends to aggregate? Or do they interact in a more specific pattern? please be more specific.

Lines 176-181 “Considering the predicted internal IRES sequence in the GLV genome, the translation of ORF1 (CP) does not initiate from the first methionine but from an internal amino acid residue [15,33]. The first amino acid residue of the CP was started from the internal Pro residue (PENIT …), according to a previous mass spectrometry analysis of the purified GLV particles (Supplementary Fig. S3)”

This sentence is not required, since the point is that the first visible residue in the structure is residue 70 or 71. It is interesting that translation of GLV CP does not begin with an AUG codon, but the sentence as written is confusing (“the first amino acid residue on the CP WAS STARTED from the internal Pro residue…”. It would be better to simply say that translation does not start with an AUG codon, but it is irrelevant for the results provided in this manuscript, and it could be rather explained in the legend to Figure S3 rather than in the main text, where it cuts the flow and leaves the reader wodering...

Lines 187-197: this paragraph does not explain whether there is a CP core that superposes well between CP-A and CP-B (with an RMSD under 1Å, for instance), and whether there is a hinge between two or more domains that leads to different inter-domain orientations in A and B , but which could be superposed as individual rigid bodies (like the “carapace”, “apical” and “dimerization” domains defined for the in the internal CP of blue tongue virus (DOI: 10.1038/26694), or whether the large RMSD is distributed evenly throughout the molecule. Also, 4.3Å and 8.9Å RMSD differences in GLV-HP and GLV-CAT are quite substantial differences, and the reason for this does not stand out in Figure S4. This requires more explanation. Also, it is important to quote the number of C-alpha atoms used in the comparison that led to the quoted RMSD. The manuscript could be much more informative by taking these comments into account.

Line 281: What is meant by “which have a non-exceptional intracellular transmission mechanism”, why is it “non-exceptional”?

line 285: “Such as budding”. this term is normally used for enveloped viruses, which bud out from a cell by incorporating a patch of the cell membrane. But in the case of non-enveloped viruses such as GLV, it is important to explain what is meant by “budding”.

Lines 398-400: “Nano differential scanning fluorimetry (nanoDSF) thermographs suggest differences in Ti (inflection temperature) values between GLV-HP and GLV-CAT particles”. Do these measurements “suggest” or show differences in Tm? If these are experimental measurements, they show that there are differences, not “suggest”.

Lines 414-417: “The structural differences between the two GLV prototypes show significant conformational differences in the C-terminal extension and the variable loop that are critical for CP–CP interactions”. Maybe it would better to say “`the comparative analysis of the two GLV prototypes reveal significant conformational differences”, rather than the redundant sentence “the structural differences reveal significant conformational differences”.

Figures:

Figure 1, left panels. The transparent surface representation does not stand out. The squared frame does a better job.

Right panels: It would be useful to label some of the residues that are drawn in sticks and appear in all the panels, for instance, R923 (which should be colored according to atom type in panel E). They should be labelled at every instance in which they are displayed, not only in some panels (for instance, D927 is only labelled in panel B, whereas it appears to be present in D and E as well). This would provide the readers with way to better compare the various panels.

In FIgure3, arrows pointing at the unique surface loops of GLV and OmRV would make it easier to see their location, rather than the transparent surfaces, which do not stand out. It would also be useful to have an overall RMSD of the superposition by for the common core of the two proteins. I presume the authors are comparing CP-A with CP-A (or CP-B with CPB) of each virus. This should be stated in the Figure legend.

Sincerely,

Félix A. Rey

Academic Editor

PLOS Pathogens

Guangxiang Luo

Section Editor

PLOS Pathogens

Michael Malim

Editor-in-Chief

PLOS Pathogens

orcid.org/0000-0002-7699-2064

The revised manuscript by Wang et al is improved with respect to the original version. The comparative study of the two GLV isolates is interesting, and the results are informative to better understand the life cycle of these dsRNA viruses. The manuscript, however, lacks clarity at certain places, some of which are specified below. Also, it is important that the manuscript be seen by a native English speaker, as the narrative is sometimes ambiguous and difficult to follow. I raise some of the issues below, as well as comments on the Figures.

Line 168-169: “the GLV-HP particles tend to interact with each other”: does this mean the purified particle preparation tends to aggregate? Or do they interact in a more specific pattern? please be more specific.

Lines 176-181 “Considering the predicted internal IRES sequence in the GLV genome, the translation of ORF1 (CP) does not initiate from the first methionine but from an internal amino acid residue [15,33]. The first amino acid residue of the CP was started from the internal Pro residue (PENIT …), according to a previous mass spectrometry analysis of the purified GLV particles (Supplementary Fig. S3)”

This sentence is not required, since the point is that the first visible residue in the structure is residue 70 or 71. It is interesting that translation of GLV CP does not begin with an AUG codon, but the sentence as written is confusing (“the first amino acid residue on the CP WAS STARTED from the internal Pro residue…”. It would be better to simply say that translation does not start with an AUG codon, but it is irrelevant for the results provided in this manuscript, and it could be rather explained in the legend to Figure S3 rather than in the main text, where it cuts the flow and leaves the reader wodering...

Lines 187-197: this paragraph does not explain whether there is a CP core that superposes well between CP-A and CP-B (with an RMSD under 1Å, for instance), and whether there is a hinge between two or more domains that leads to different inter-domain orientations in A and B , but which could be superposed as individual rigid bodies (like the “carapace”, “apical” and “dimerization” domains defined for the in the internal CP of blue tongue virus (DOI: 10.1038/26694), or whether the large RMSD is distributed evenly throughout the molecule. Also, 4.3Å and 8.9Å RMSD differences in GLV-HP and GLV-CAT are quite substantial differences, and the reason for this does not stand out in Figure S4. This requires more explanation. Also, it is important to quote the number of C-alpha atoms used in the comparison that led to the quoted RMSD. The manuscript could be much more informative by taking these comments into account.

Line 281: What is meant by “which have a non-exceptional intracellular transmission mechanism”, why is it “non-exceptional”?

line 285: “Such as budding”. this term is normally used for enveloped viruses, which bud out from a cell by incorporating a patch of the cell membrane. But in the case of non-enveloped viruses such as GLV, it is important to explain what is meant by “budding”.

Lines 398-400: “Nano differential scanning fluorimetry (nanoDSF) thermographs suggest differences in Ti (inflection temperature) values between GLV-HP and GLV-CAT particles”. Do these measurements “suggest” or show differences in Tm? If these are experimental measurements, they show that there are differences, not “suggest”.

Lines 414-417: “The structural differences between the two GLV prototypes show significant conformational differences in the C-terminal extension and the variable loop that are critical for CP–CP interactions”. Maybe it would better to say “`the comparative analysis of the two GLV prototypes reveal significant conformational differences”, rather than the redundant sentence “the structural differences reveal significant conformational differences”.

Figures:

Figure 1, left panels. The transparent surface representation does not stand out. The squared frame does a better job.

Right panels: It would be useful to label some of the residues that are drawn in sticks and appear in all the panels, for instance, R923 (which should be colored according to atom type in panel E). They should be labelled at every instance in which they are displayed, not only in some panels (for instance, D927 is only labelled in panel B, whereas it appears to be present in D and E as well). This would provide the readers with way to better compare the various panels.

In FIgure3, arrows pointing at the unique surface loops of GLV and OmRV would make it easier to see their location, rather than the transparent surfaces, which do not stand out. It would also be useful to have an overall RMSD of the superposition by for the common core of the two proteins. I presume the authors are comparing CP-A with CP-A (or CP-B with CPB) of each virus. This should be stated in the Figure legend.

Reviewer Comments (if any, and for reference):

Figure Files:

Data Requirements:

Reproducibility:

References:

---

## [Editor Report · Decision Letter 2]

21 Mar 2024

Dear Dr. Okamoto,

We are pleased to inform you that your manuscript 'High-resolution comparative atomic structures of two Giardiavirus prototypes infecting G. duodenalis parasite' has been provisionally accepted for publication in PLOS Pathogens.

Best regards,

Félix A. Rey

Academic Editor

PLOS Pathogens

Guangxiang Luo

Section Editor

PLOS Pathogens

Michael Malim

Editor-in-Chief

PLOS Pathogens

orcid.org/0000-0002-7699-2064
---

## [Editor Report · Acceptance letter]

4 Apr 2024

Dear Dr. Okamoto,

We are delighted to inform you that your manuscript, "High-resolution comparative atomic structures of two Giardiavirus prototypes infecting G. duodenalis parasite," has been formally accepted for publication in PLOS Pathogens.

Best regards,

Michael Malim

Editor-in-Chief

PLOS Pathogens

orcid.org/0000-0002-7699-2064